# Hypoplastic and Congenital Absence of Coronary Arteries and Its Correlation with Clinical Implications of Cardiac Circulation: A Systematic Review and Meta-Analysis

**DOI:** 10.3390/jcm13113085

**Published:** 2024-05-24

**Authors:** Alejandro Bruna-Mejias, Javiera del Villar-Valdebenito, Camila Roman, Catalina Alcaíno-Adasme, Walter Sepulveda-Loyola, Mathias Orellana-Donoso, Pablo Nova-Baeza, Alejandra Suazo-Santibañez, Alvaro Becerra-Farfan, Juan Sanchis-Gimeno, Juan José Valenzuela-Fuenzalida

**Affiliations:** 1Departamento de Ciencias y Geografía, Facultad de Ciencias Naturales y Exactas, Universidad de Playa Ancha, Valparaiso 2360072, Chile; alejandro.bruna@unab.cl; 2Facultad de Medicina, Universidad Mayor, Santiago 8580745, Chile; javiera.delvillar21@gmail.com; 3Department of Morphology, Faculty of Medicine, Universidad Andres Bello, Santiago 7500735, Chile; camilaroman.cfrn@gmail.com (C.R.); cata.alcaino.adasme@gmail.com (C.A.-A.); pablo.nova@usach.cl (P.N.-B.); 4Faculty of Health Sciences, Universidad de las Américas, Santiago 8320000, Chile; walterkine2014@gmail.com (W.S.-L.); alej.suazo@gmail.com (A.S.-S.); 5Department of Morphological Sciences, Faculty of Medicine and Science, Universidad San Sebastián, Santiago 8420524, Chile; mathor94@gmail.com; 6Escuela de Medicina, Universidad Finis Terrae, Santiago 7501015, Chile; 7Departamento de Ciencias Química y Biológicas, Facultad de Ciencias de la Salud, Universidad Bernardo O’Higgins, Santiago 8320000, Chile; alvaro.becerra@ubo.cl; 8GIAVAL Research Group, Department of Anatomy and Human Embryology, Faculty of Medicine, University of Valencia, 46001 Valencia, Spain; juan.sanchis@uv.es

**Keywords:** anatomy coronary artery, hypoplastic coronary artery, agenesia coronary artery, clinical anatomy, variations anatomical

## Abstract

**Background:** Coronary arteries originate from the first portion of the aorta, emerging from the right and left aortic sinuses. They traverse through the subepicardium and coronary sulcus to supply the myocardium during diastolic function. The objective of this review was to understand how the hypoplasia and agenesis of the coronary arteries are associated with cardiac pathologies. **Methods:** The databases Medline, Scopus, Web of Science, Google Scholar, CINAHL, and LILACS were researched until January 2024. An assurance tool for anatomical studies (AQUA) was used to evaluate methodological quality. The pooled prevalence was estimated using a random effects model. **Results:** A total of three studies met the established selection criteria for inclusion in this meta-analysis. The prevalence of coronary artery variants was 3% (CI = 2% to 8%), with a heterogeneity of 77%. The other studies were analyzed descriptively, along with their respective clinical considerations in the presence of the variant. **Conclusions:** Hypoplasia and the congenital absence of the coronary arteries are often incidental findings and understanding these variants is crucial to prevent misdiagnosis. Additionally, it is essential to exercise caution when considering surgical management for hearts with these variants.

## 1. Introduction

Coronary arteries originate from the first portion of the aorta, emerging from the right and left aortic sinuses [1]. The right coronary artery (RCA) has an anterior to posterior path through the coronary sulcus, where it originates its branches, highlighting, due to its importance, the sinus node artery, right marginal artery, and the posterior descending artery. Altogether, these branches provide the blood supply to most of the heart tissue [1].

The left main coronary artery (LMCA) has a short path in the coronary sulcus. It originates, almost immediately, the left anterior descending artery (LDA) and the left circumflex artery (LCx). LCx arteries follow the coronary sulcus on the left side and originate the left marginal artery in charge of the blood supply of the left ventricle (Loukas et al., 2013 [1]). However, coronary arteries have normal anatomical variations, especially in the dominance of the heart blood supply. The most common dominance is the RCA (81.6%), followed by LCx (12.2%), and the least common is the co-dominance (6.2%) in heart irrigation [2]. These variations between subjects take relevance when presenting coronary artery disease, where the blood supply can be affected in the dominant vessel.

The embryological development of the coronary system is essential when mentioning anomalies such as hypoplasia and the absence of the coronary arteries, since most of them have their origin in intrauterine life. Prior to coronary embryogenesis, the heart is a thin muscular layer that oxygenates through diffusion. After its thickening, it requires the development of a vascular plexus that will mature through anastomosing with the aorta and provide the myocardial tissue with blood, oxygen, and nutrients. It is imperative to mention the signaling pathways involved in the process of the normal coronary development, such as Tbx5, Tbx18 and Vegf-A/Vegfr2, as some studies show that through the knockout of the cell or molecules involved in the pathways, they can produce relevant anomalies in the coronary arteries such as hypoplasia or congenital absence. Coronary artery development is a complex vasculogenic process that begins shortly after the cardiac loop. Coronary vasculogenesis is regulated by the myocardium but depends spatially and temporally on the epicardium and its precursor, the proepicardial organ, for the supply of coronary vascular progenitor cells. Changes in the disposition of cardiac precursors may be the cause of coronary alterations such as hypoplasia [3,4,5].

The absence of the coronary artery is a rare congenital defect. According to an angiographic imaging study, only 0.056% of the 30,230 subjects presented a single coronary artery, where 65% originated from the right sinus of the Valsalva and 35% from the left side [4]. It can be said that the congenital absence of the LMCA is more common. However, it is worth mentioning that the absence of LMCA branches, such as LCx, is far more common, where the complete congenital absence of the LMCA has a much lower prevalence at 0.024% [5]. In general, the hypoplasia or congenital absence of a coronary artery does not have clinical relevance, being an asymptomatic and incidental finding. The clinical relevance of these anatomical defects manifests when coexisting with coronary artery disease, where the single coronary artery cannot provide the blood supply of its own territory and the territory of the missing artery, resulting in myocardial ischemia due to a lack of oxygen and nutrients. Coronary artery disease can be the cause of different ischemia-like symptoms such as exertion angina, syncope, or episodic tachycardia in patients with congenital coronary artery defects. A study reports that 15% of the patients with the congenital absence of a coronary artery have ischemic damage due to its condition [6]. The most serious consequence of the congenital absence of a coronary artery is sudden death due to the complete occlusion of the super-dominant artery. Coronary anomalies are the second most common cause of sudden death in healthy-looking young people [7].

Therefore, this review aims to analyze and characterize congenital coronary artery diseases such as the hypoplasia and absence of coronary arteries, along with highlighting their importance for differential diagnosis in patients with ischemia-like symptoms and its cardiologic repercussions.

## 2. Methods

### 2.1. Protocol and Registration

This systematic review and meta-analysis adhered to the guidelines outlined in the Preferred Reporting Items for Systematic Reviews and Meta-Analyses (PRISMA) statement [8]. The registration number in the International Prospective Register of Systematic Reviews (PROSPERO) is CRD42024520734.

### 2.2. Eligibility Criteria

Studies included in this review were selected using the following criteria:

(1) Population: Samples from cadaveric dissections and live images of hypoplastic coronary artery diseases (HCAD); (2) Results: Examination of the prevalence of CA variants and their correlation with hearth pathologies, with a specific focus on abnormalities of the CA. In addition, anatomical variants were classified and described based on normal anatomy and classifications and descriptions proposed in the literature; (3) Studies: Research articles and case reports involving only human samples, and original research published in English in peer-reviewed journals and indexed in the reviewed databases. Excluded from consideration were letters to the editor.

### 2.3. Electronic Search

The systematic research was performed using the subsequent databases: MEDLINE (via PubMed), Web of Science, Google Scholar, the Cumulative Index to Nursing and Allied Health Literature (CINAHL), Scopus, and the Latin American and the Caribbean Literature in Health Sciences (LILACS) from inception until January 2024. The search strategy involved a combination of the following terms: “Anatomy coronary artery” (no mesh), “Hypoplastic coronary artery” (no mesh), “agenesis coronary artery” (no mesh), “clinical anatomy” (no mesh), “irrigation heart” (no mesh), “congenital absence of coronary arteries” (no mesh), and “variations anatomical” (no mesh), using the Boolean connectors AND, OR, and NOT. The search strategies for each database are available in the Appendix A.

### 2.4. Study Selection

Two authors (CAA and JDV) independently screened the titles and abstracts of references retrieved from the searches. The full text was obtained for references that either author considered potentially relevant. We involved a third reviewer (JV) if consensus could not be reached. The inter-evaluator validity was assessed using the Kappa index, yielding a value of 0.68.

### 2.5. Data Collection Process

Two authors (MO and AQ) independently extracted data on the outcomes of each study. The following data were extracted from the original reports: (i) Authors and year of publication, (ii) Country, (iii) Age and sex, (iv) Prevalence, (v) Clinical history, (vi) Circumstances, (vii) HCAD (Hypoplasia of the Coronary Artery Disease), (viii) Congenital absence of the Coronary Artery, (ix) Aberrant origin, (x) Other, and (xi) Clinical implications.

### 2.6. Assessment of the Methodological Quality of the Included Studies

The quality assessment was performed using the methodological quality assurance tool for anatomical studies (AQUA) proposed by the International Evidence-Based Anatomy Working Group (IEBA) [9]. Data extraction and quality assessment were independently performed by two reviewers (JJV and CR). We involved a third reviewer (JSG) if a consensus could not be reached. The agreement rate between the reviewers was calculated using Kappa statistics 0.80.

### 2.7. Statistical Methods

The data extracted from the meta-analysis were interpreted by calculating the VAH prevalence using JAMOVI software (Version 1.12) [accessed January 2024]. The DerSimonian–Laird model with a Freeman–Tukey double arcsine transformation was used to combine the summary data. In addition, a random effects model was used, because the VAH prevalence data were highly heterogeneous. The degree of heterogeneity between included studies was assessed using the chi^2^ test and the heterogeneity (I^2^) statistic. For the chi^2^ test, the *p* value proposed by the Cochrane collaboration was considered significant at 0.10. Values of the I^2^ statistic were interpreted with a 95% confidence interval [CI] in the following way: 0–40% might not be important, 30–60% might indicate moderate heterogeneity, 50–90% might represent substantial heterogeneity, and 75–100% could represent a significant amount of heterogeneity.

## 3. Results

### 3.1. Included Articles

A total of 195 articles from different databases met the criteria and search terms established by the research team. The filter was applied to the titles and/or abstracts of the articles in the consulted databases, and the primary criterion for eliminating duplicates was used. Subsequently, 112 full-text articles were evaluated for their eligibility for inclusion in this meta-analysis and systematic review. Eighty-eight studies were excluded due to discrepancies between their primary and secondary results compared to those of this review, or because they failed to meet the established criteria for good data extraction. Therefore, 39 articles [10,11,12,13,14,15,16,17,18,19,20,21,22,23,24,25,26,27,28,29,30,31,32,33,34,35,36,37,38,39,40,41,42,43,44,45,46,47,48] were selected for this study (*n* = 140,137) (including patients, images, and cadavers) (Figure 1).

### 3.2. Characteristics of the Studies and Population

Of the 39 studies included in this review (as listed in Table 1 and Table 2), 19 articles are from Asia [11,12,13,19,20,21,23,27,33,35,36,37,39,41,43,45,46,47,48], 10 are from Europe [13,16,17,22,26,30,31,32,34,40], 8 are from North America [10,15,18,24,25,29,38,44], 1 is from South America [28], and 1 is from Oceania [42]. Therefore, the total number of subjects is 140,137, among whom 811 presented coronary artery variants, with an average age of diagnosis of 41.4 years.

### 3.3. Variant Description

#### 3.3.1. Right Coronary Artery Variants

The right coronary artery (RCA) usually emerges from the first portion of the aorta. The development of the coronary arteries begins between the sixth and seventh weeks of intrauterine life; this period is where congenital anomalies, such as the hypoplasia or absence of coronary arteries occur. The variants and the repercussions they may cause in the function of the heart were analyzed, and featured in 16 articles:

The congenital absence of the RCA was found in nine articles [10,12,16,19,22,26,27,30,45], with a total of ten study subjects, characterizing this abnormality as the complete absence of the RCA or one of its branches. This variation results in insufficient blood supply in the RCA territory of the heart, which may lead to myocardial ischemia during exertion. To compensate for this deficiency, the heart may supply this area by extending the irrigation territory of the left main coronary artery (LMCA) or its branches to ensure proper blood supply to the heart.

The super dominance of the circumflex artery (LCx) was found, mentioned in three studies featuring three subjects [10,22,27], followed by LMCA super dominance in two studies with two subjects [22,45]. For this reason, this variant is mainly asymptomatic and only exhibits symptoms during exertion.

Another coronary variant found was hypoplastic RCA, with five studies showing that 563 subjects out of a total of 5957 have this anomaly [17,24,38,39,42]. Right hypoplastic coronary artery disease refers to the partial absence or incomplete development of the RCA or its branches. This anomaly has the same consequences as the absence of the vessel, such as ischemia due to insufficient blood supply to the heart during exertion. In one study, hypoplastic RCA was found along with other anatomical pathologies such as single RCA, split RCA, and the abnormal origin of the RCA [11]. Although hypoplasia is believed to be a congenital disorder, we found one study that identified, through genomic sequencing, a NOTCH1 c.1023CA mutation that can cause H-RCA [37].

#### 3.3.2. Left Coronary Artery Variants

The left main coronary artery (LMCA) also arises from the aorta, having a short course before the circumflex artery (LCx) and the left anterior descending artery (LAD) emerge. The congenital anomalies, such as congenital absence and hypoplastic LMCA or its branches, occur during the gestation period, and their effects on blood supply and the function of the heart were analyzed.

The complete absence of the LMCA was found in eight studies with 632 subjects out of a total of 7507 [11,15,26,30,33,40,46,48]. The authors concluded that the congenital absence of the LMCA tends to be asymptomatic and its finding is often incidental. To compensate for the deficiency it may cause, the heart can supply this area through the overdevelopment of the RCA, causing dominancy found in four articles [13,15,18,23], the super dominance of RCA, found in seven articles [20,28,31,34,36,40,43,47], and/or the aberrant origin of LAD or LCx from the RCA found in four articles [11,15,28,44]. These anatomical changes may ensure the correct blood supply to the myocardium. However, one study highlights the relationship between the congenital absence of the LMCA and H-RCA [11]. In both cases, the symptoms manifest when the heart is overloaded and the blood supply is insufficient, ranging from ischemia to myocardial infarction, ventricular fibrillation, or even death [33,40,48]. The congenital absence of the LMCA can also be related to the congenital absence of LAD, a correlation that can also cause ischemia-like symptoms during exertion [15].

The hypoplasia of the LMCA presented fewer incidences, being studied in only four studies with four subjects [14,17,25,35]. Hypoplastic left coronary artery (H-LMCA) refers to the underdevelopment of the LMCA, causing an insufficient blood supply to the heart that can or cannot be compensated by the right coronary system or the aberrant origin of the LCx or LAD. This condition may have symptoms like acute myocarditis, cardiomyopathy, myocardial ischemia, or cardiac failure. Thus, it becomes important to consider these variants when dealing with patients having cardiovascular complications [15].

The most common vessel with these anomalies is the circumflex artery (LCx), found in eleven studies with 62 subjects [18,20,21,23,28,31,34,36,41,47].

The congenital absence of LCx is often an incidental finding [31] and may even be benign [18], sometimes being discovered post-mortem [34]. The absence itself usually lacks major significance due to the compensatory blood supply provided by the overdevelopment of the RCA, LMCA, or LAD, which can adequately irrigate the affected area up to a certain extent. Its clinical importance lies in its association with other pathologies such as atherosclerosis [36], which can lead to ischemic-like symptoms during exertion [23,31], syncope, or even sudden death (Oliveira, 2015 [28]). Additionally, we found two studies featuring two subjects with situs inversus and a congenital absence of the LCx [41,43]. In one case, the patient was asymptomatic, and the absence was incidental [41], while in the other case, the patient experienced angina-like symptoms due to ischemia caused by the congenital absence of the LCx, along with significant stenosis on the LAD [43]. This situation compromises the anatomical compensation of the heart vessels in providing adequate blood supply.

Furthermore, LMCA branches such as LCx and LAD can also be hypoplastic. This anatomical variation was found in four studies with a total of five subjects [13,24,32,35]. The underdevelopment of these vessels may or may not be compensated by the right coronary system. When there is no compensation, the blood supply is insufficient during exertion, leading to symptoms such as dyspnea, palpitations, syncope [24], myocardial ischemia, or even sudden death [13]. A hypoplastic LCx artery may be due to a deficiency of NOS3 during intrauterine life according to Riede et al., 2013 [32].

One of the most common classifications is that of Yamanaka et al., which is characterized below. This classification is based on the location of the coronary artery ostium (right or left) and its direction, and its relationship with other vessels. In single-vessel coronary artery type I, the single artery has a normal course, and collateral vessels compensate for the absent coronary artery. In type II, the anomalous coronary artery arises near the opposite coronary artery, passes through the heart base, and its relation to large vessels may vary: anterior course in relation to pulmonary and aortic arteries (subtype A), between large vessels of the base (subtype B), posterior to the large vessels of the base (subtype P), and combinations of these (C). In type III, the proximal segment of the right coronary artery originates the anterior descending artery and the circumflex branch separately (refer to Figure 2 and Figure 3).

### 3.4. Prevalence

To calculate the prevalence of variant coronary arteries in the studies included in this review, four proportion forest plots were created. Three studies [11,39,44] were included for the prevalence of congenital absence or hypoplastic coronary artery, presenting a prevalence of 3% (CI from 2% to 8%). The heterogeneity for the included sample was 94% (Figure 4). However, the funnel plot did not apply to this sample due to the low number of studies included.

### 3.5. Risk of Bias of Included Articles

Three studies met the criteria for evaluation using the AQUA Checklist for Anatomical Studies tool, which analyzes bias in five domains. In all five domains provided by the AQUA table, the three included studies demonstrated a low risk of bias and were analyzed comprehensively (refer to Figure 5).

For the analysis of studies with case report methodology, the JBI tool was utilized to evaluate the risk of bias in this type of study. A total of 36 studies [10,12,38,40,41,42,43,45,46,47,48] were analyzed across the eight domains of this bias assessment tool (refer to Figure 6). The majority exhibited a low risk of bias. Specifically, eleven studies showed a high risk of bias in Domain 7, nine studies presented a high risk of bias in Domain 8, and ten studies demonstrated a high risk of bias in Domain 8.

### 3.6. Clinical Considerations

For the reported clinical correlation, we have found some clinical presentations reported by the articles included in the review.

Most of the articles concluded that the congenital absence or hypoplasia of CA on its own lacks clinical relevance, being a relatively benign and asymptomatic finding [16,18,19,22,26,33,34,48] due to the overdevelopment of the other vessels [22]. However, it can be relevant when the absence/hypoplasia or super dominance of the other coronary arteries (or its branches) coexist with another pathology that exacerbates the insufficient blood supply to the myocardium, or when the lifestyle of the patient requires a high oxygen demand as an athlete [10,26,36]. Eight studies [10,12,17,22,27,36,43,45] reported that the coexistence of these variants and atherosclerotic disease can lead to an insufficient blood supply to the myocardium due to the restriction in the super-dominant vessel that replaces the irrigation, resulting in ischemia-like symptoms in exertion such as angina pectoris or even myocardial infarction with complete (STEMI) [12,27] or incomplete (NSTEMI) [10,17] vessel occlusion.

The most common symptoms that lead the patients to seek medical attention were described in nine studies [15,17,23,24,28,31,38,40,48] which included: ischemia-like symptoms [28,40,48], angina pectoris in exertion [15,17,23,31], syncope [24], dyspnea [38], palpitations [24], and bradycardia [38].

A chronic insufficient blood supply of the myocardium can lead to an ischemic disease, described in four studies [17,24,38,42], characterized by ventricular dysfunction or ventricular dilatation [24], unstable angina [38], acute myocardial infarction, and death [17,38,42].

Other clinical manifestations were less common, but worth mentioning due to their relevance in the prevention or treatment of the patient:

First, one study [10] has proposed the possible early and abnormal degeneration of the sinoatrial nodule or atrioventricular nodule due to the congenital absence of the RCA, which is normally responsible for the blood supply of these structures. This early degeneration can lead to a nodal conduction dysfunction, resulting in arrhythmias or ischemia [10].

A study [11] found that 36 of 46 patients have hypoplastic RCA related to the congenital absence of the LMCA, which can result in an increased risk for myocardial ischemia [11].

On the other hand, another study [10] said that the increased blood flow to the remaining CA, which supplies the missing vessel irrigation territory, can also increase its diameter, causing compression ischemia in nearby structures as well as the endothelial dysfunction and/or atherosclerosis of the CA [10].

Finally, it is critical to be aware of these anatomical variations (congenital absence or hypoplasia) as a patient with ischemia-like symptoms or cardiac failure may have them. Hence, it is important to consider the possible presence of these variants, and to perform a proper diagnosis and treatment [14]. In addition, even myocarditis and cardiomyopathy [14] can manifest when these variants are present, even leading to sudden death [13,28].

It is essential to emphasize that it is possible and common to find more than one coronary variant in the same patient; in fact, the articles that mention a unique variant are fewer [10,11,24]. Considering this, as these anatomical anomalies may mimic many cardiac diseases and CAD, it becomes critical to perform, as mentioned before, a correct and profound analysis of the exams and also to conduct proper tests that allow the medical team to detect these anomalies, such as selective coronary angiography (SCA), coronary angiogram, or computerized tomography coronary angiography (CTA), to avoid overlooking a coronary artery (CA) variant [12,13]. It is also recommended to corroborate the results with MDCT [12].

Many of the reviewed articles recommend treating HCAD by implanting an ICD to improve the quality of life. However, this may not be effective for all patients because its success depends on the severity of the blood supply alteration caused by the variants.

## 4. Discussion

In this systematic review and meta-analysis, coronary artery variants have been studied, primarily focusing on HCA as well as the congenital absence of the coronary artery and its branches. These variants, uncommon in the population, exhibit considerable variability in their description and regions of occurrence. The severity of this condition is contingent upon several factors, including the degree of narrowing in the HCA, the presence of single or multiple coronary artery variants, compromised areas, the presence of underlying chronic heart disease, and the lifestyle of the individual, regarding its requirement for high oxygen demand. These abnormalities may go unnoticed; therefore, their detection is mostly incidental during surgeries or imaging exams. Furthermore, once HCAD is diagnosed, it needs to be treated as soon as possible, as it can cause severe pathologies leading to death. Previous reviews have not found detailed analyses of these variants or studies that directly associate them with clinical considerations. This study is the first in this line of research and is innovative; it may be important to consider during clinical surgical management.

There is an evident prevalence of studies conducted in Asia and Europe, rather than in the Americas and Oceania. However, among the 140,137 subjects included in the study, 126,602 (90.34%) are from North America, 6019 (4.29%) from Asia, and 7514 (2.30%) from Europe, with only one subject each from South America and Oceania (0.0007%).

Regarding subjects presenting coronary artery abnormalities, there is a clear ethnic prevalence. Out of the 811 subjects with coronary artery variants, the majority (639; 78.79%) are from Europe, followed by Asia with 159 (19.60%), and a minority from North America with 11 (1.35%). On the other hand, South America and Oceania each have one subject (0.12%) with a coronary abnormality [49].

Additionally, sex distribution was examined. Among the subjects, 35,049 were identified as female and 99,135 as male, accounting for 25.01% female and 70.74% male subjects, respectively. However, sex was not specified in 93 articles, encompassing 5953 subjects (4.27%). Therefore, there is a prevalence ratio of 2.8 to 1 between males and females included in the study. Nonetheless, insufficient information about the sex of the patients precludes determining the prevalence of these variants between genders.

Regarding the prevalence of this variant, only three studies met the criteria for random sample analysis, so the prevalence reported may not reflect the true prevalence. It is also important to note that this type of AC variant is most commonly identified symptomatically in patients at various life stages or during surgical procedures on adjacent structures. Despite the majority of the included studies being case studies, the risk of bias was low, indicating methodological rigor.

Understanding these anatomical variations in the coronary arteries is crucial in clinical practice, particularly for the differential diagnosis of cardiovascular diseases and understanding the implications of results and repercussions of procedures and surgeries. The objective is to identify these anomalies to prevent complications and ensure successful recovery or an improved quality of life [50].

Concerning the symptoms, most studies showed that the anatomical anomaly tends to be asymptomatic when there are no other pathologies. This means that it is often an incidental and benign finding in patients with healthy lifestyles. However, it can be symptomatic by itself if the blood supply is insufficient to the heart or the nodules, causing ischemia or arrhythmias.

The real clinical value of knowing if a patient has hypoplasia or a congenital absence of the coronary arteries lies in the coexistence with other diseases, such as atherosclerosis, hypertension, diabetes, metabolic syndrome, or others that are less frequent, like acute cardiac failure or ventricular hypertrophy. This can lead to ischemia-like symptoms, angina pectoris, dyspnea, an increased risk of myocardial infarction, or even sudden death, among other complications. Most of the mentioned complications can be preventable with surgical procedures, lifestyle changes, or through treating the base pathology.

It is also relevant to be aware of the super-dominant vessel and the preventable repercussions of not treating it, as it can increase in diameter and cause the compression of nearby structures, leading to endothelial dysfunction or ischemia in previously healthy tissue.

Finally, it is important to always think of the slight chance of a congenital absence or hypoplasia of the coronary artery when dealing with patients with cardiac failure, myocarditis, or cardiomyopathy, as these conditions are differential diagnoses to a coronary artery variation and must be discarded to treat the patient and their condition in the best way possible, ensuring a correct diagnosis and therefore the corresponding treatment.

## 5. Limitations

This review was limited by the publication and authorship bias of the included studies. First, studies with different results that were in the non-indexed literature in the selected databases may have been excluded. Second, there could be limitations in the sensitivity and specificity of the searches. Finally, the authors personally selected the articles. All of this increases the probability of excluding potential cases from countries outside of Asia and North America that are not being reported in the scientific community.

## 6. Conclusions

In this systematic review, we have studied some congenital anomalies that affect CA, which have a low prevalence and are most often asymptomatic. The congenital absence of LMCA plus hypoplastic RCA are associated with an increased risk of ischemia, while myocarditis and cardiomyopathy are more likely to manifest in the presence of congenital coronary arteries and are associated with an increased risk of sudden death. It is important that cardiovascular doctors and surgeons recognize the variants of this anomaly for the adequate management of asymptomatic patients to be diagnosed early to have less probability of future complications. Finally, we believe that new studies could better study the relationship between these conditions and different cardiovascular pathologies.

## Figures and Tables

**Figure 1 jcm-13-03085-f001:**
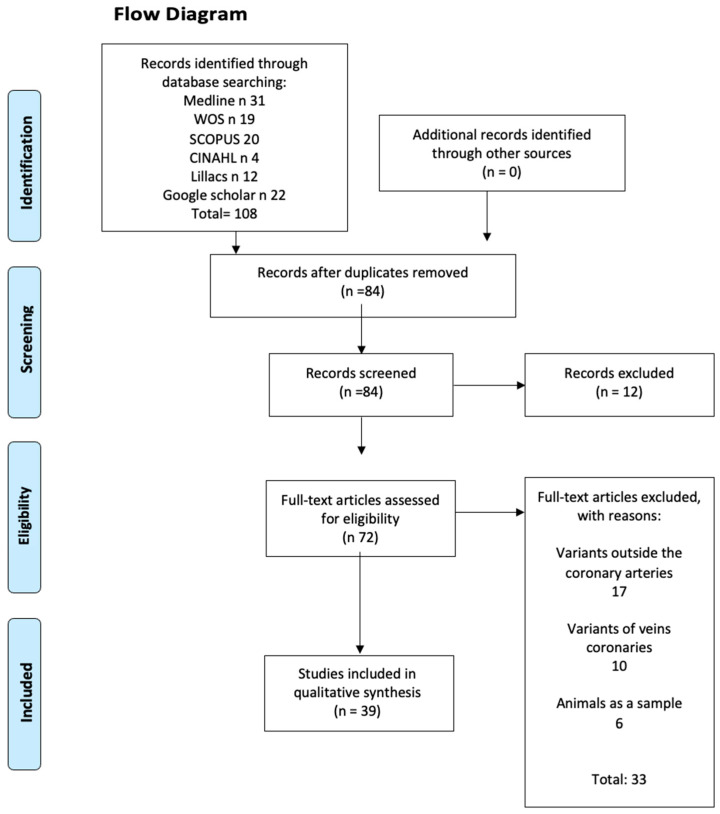
Flow chart CA.

**Figure 2 jcm-13-03085-f002:**
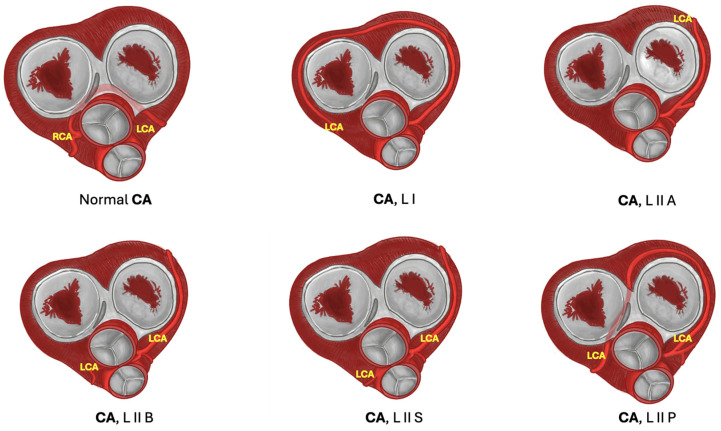
Variants of left coronary artery. In the diagram, LCA variants are shown. Normal CA: Normal anatomical course of both CA. L I: Single LCA passes slightly superior to its normal course and extends its territory to compensate RCA absence. L II A: LCA passes superior to its normal course, laterally to the ascending aorta and pulmonary trunk, and surrounding left atrioventricular valve. L II B: LCA passes between the ascending aorta and pulmonary trunk. L II S: LCA crosses the pulmonary trunk to the right portion of the heart. L II P: LCA passes across the right atrioventricular valve in order to move to the right side of the heart. **Abbreviations:** RCA: right coronary artery, LCA left coronary artery.

**Figure 3 jcm-13-03085-f003:**
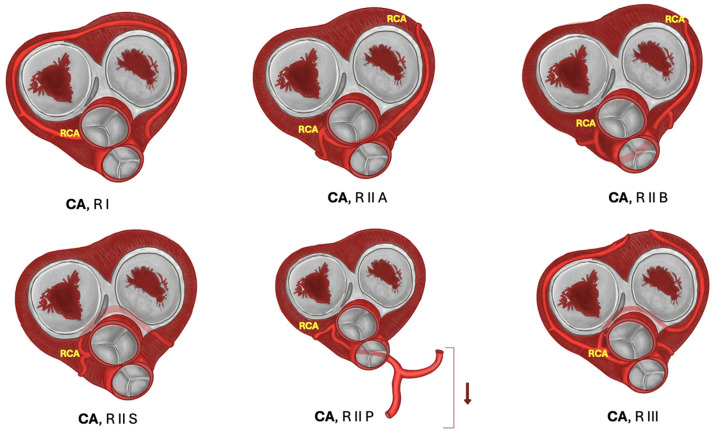
Variants of right coronary artery. In the diagram, single right coronary artery variants are shown; RCA courses slightly superior to its normal course to compensate LCA absence. R I: Single RCA variant passes superior to the normal course, and extends its territory in order to compensate LCA absence. R II A: RCA passes between the pulmonary trunk and the ascending aorta. R II B: RCA courses anterior to the pulmonary trunk and ascending aorta. R II S: RCA passes behind the ascending aorta, in close relation to right and left atrioventricular valves. R II P: RCA courses behind the pulmonary trunk and descends to the inferior portion of the heart. R III: LAD and LCx are arising separately from RCA. **Abbreviations:** RCA: right coronary artery.

**Figure 4 jcm-13-03085-f004:**
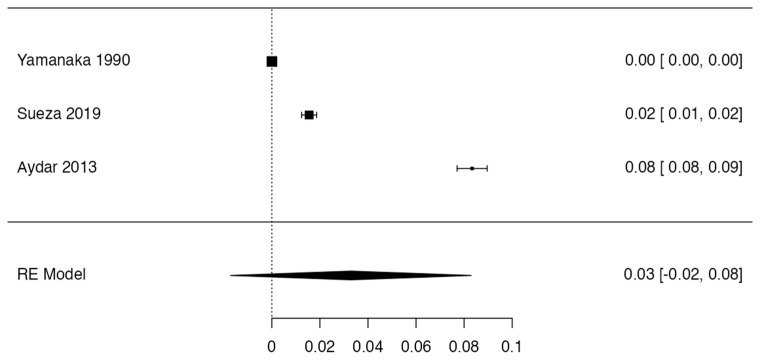
Forest plot prevalence CA.

**Figure 5 jcm-13-03085-f005:**
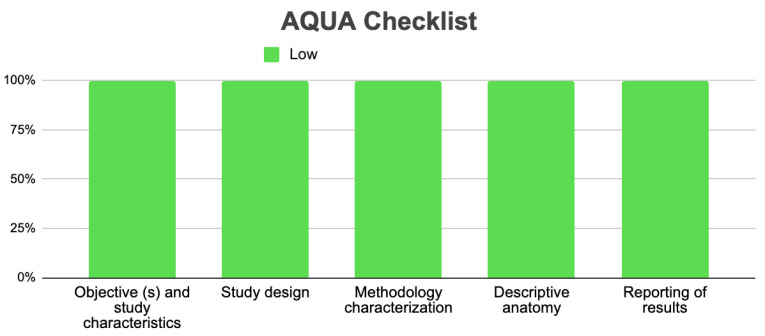
Risk of bias included studies.

**Figure 6 jcm-13-03085-f006:**
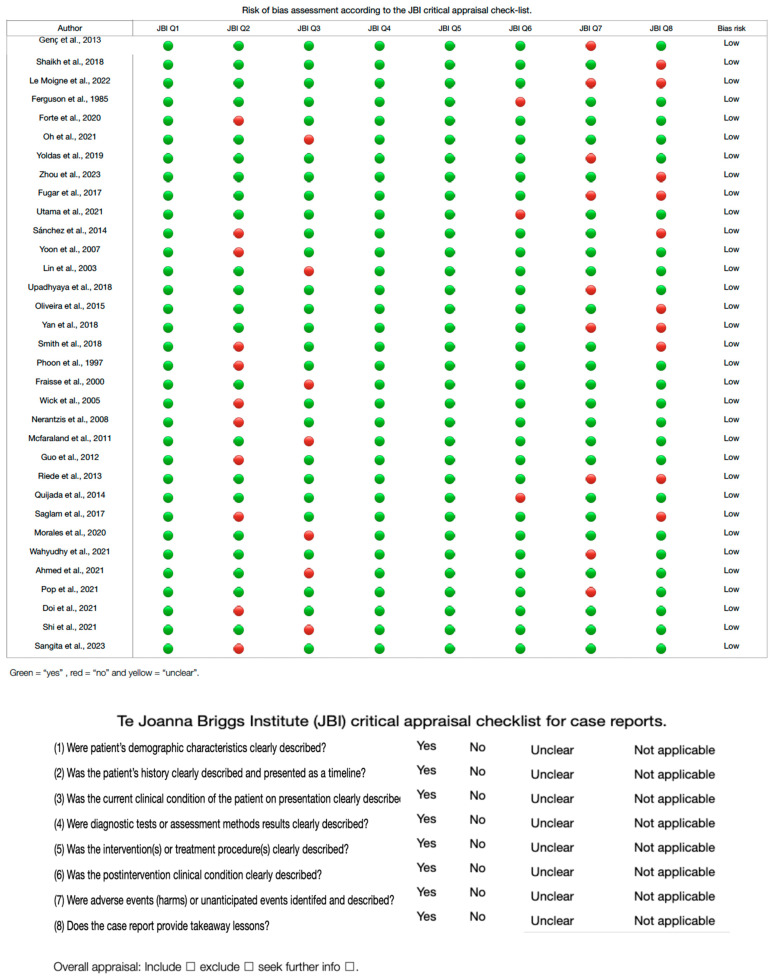
Risk of bias study of case.

**Table 1 jcm-13-03085-t001:** Geographic distribution with variants of the CA.

Geographical Region	Studies	Total *n*	Alterations Coronary Arteries
Asia	15	6019	159
Europe	14	7514	639
North of America	8	126,602	11
South of America	1	1	1
Oceania	1	1	1
Africa	0	0	0
	39	140,137	811

**Table 2 jcm-13-03085-t002:** Characteristics of the included studies.

Author/Year	G. Region	Age/Sex	F. I.	Clinical History	Symptoms	Symptoms (Yes or Not)	Circumstances	HCAD	Congenital Absence CA	Aberrant Origin	Other	Clinical Implications
Ahmed et al., 2021.	USA	66/F	1/1	HTN and HLD.	Retrosternal discomfort, shortness of breath over 3 to 4 weeks.	Yes	Clinical attention.	NA.	RCA.	NA.	Dominant LCx.	Early degeneration of SA and AV nodes may occur due to limited blood supply, potentially leading to arrhythmias or ischemia. Overexertion of LMCA can cause ischemia, therefore, further interventions may be needed in these cases.
Aydar et al., 2013	Turkey	18-102/B	625/7500	Not documented.	Asymptomatic.	Not	Clinical study.	RCA.	LMCA.	LCx, LAD.	Single RCA, Split RCA, Myocardial Bridge, Fistula.	HRCA may have an association with congenital absence of LMCA, according to this study.
Chen et al., 2020	China	54;67/B	2/2	Healthy; HTN and T2DM.	Angina and dyspnea.	Yes	Clinical attention.	NA.	RCA.	NA.	Dominant LAD.	Coronary angiography is a gold standard exam for diagnosing CA variants, but it is also recommended to use MDCT and CTA exams in order to diagnose properly.
Doi et al., 2021	Japan	8/M	1/1	Acute heart failure at 10-month-old, with vomiting and poor feeding.	Symptoms of acute coronary syndrome.	Yes	Clinical attention, misidentification of HCAD during aortography.	LMCA, LAD, LCx.	NA.	NA.	NA.	HCAD can resemble acute myocarditis or cardiomyopathy. In these cases, activity restriction is needed, close symptoms, scan monitoring, and potential ICD placement. It is important to review meticulously the exams, as HCA may be misidentified.
Ferguson et al., 1985	USA	30/F	1/1	FH for myocardial infarction.	Moderated angina over 4 years. Severe angina pain due to exertion.	Yes	Clinical attention.	NA.	LMCA, LAD.	LCx arising from RCA.	Dominant RCA.	Chest pain was not related to cardiac insufficiency according to the authors. However, considering actual information, the pain was possibly related to CA variants, causing an insufficient blood supply to the heart during exertion.
Forte et al., 2020	Italy	45/F	1/1	Exercise test suggestive of myocardial ischemia.	Asymptomatic.	Not	Incidental finding.	NA.	RCA.	NA.	NA.	Single CA disease can lead to myocardial ischemia, ventricular fibrillation, or other issues related to insufficient blood supply. This is not the case for this patient.
Fraisse et al., 2000	France	11/M	1/1	Not documented.	Angina due to exertion, myocardial infarction.	Yes	Clinical attention, ICU.	RCA, LMCA.	NA.	NA.	NA.	The patient was treated with diltiazem to continue and was discharged. Physical activity restrictions and potential ICD placement are needed in this case.
Fugar et al., 2017	USA	46/M	1/1	No significant medical history.	Loss of consciousness after a mechanical fall. There was a sinus rhythm with frequent premature ventricular contractions (PVCs) shown in the EKG.	Yes	Clinical attention in the emergency department.	NA.	LCx.	NA.	Dominant RCA.	Benign finding according to the authors. Congenital absence of LCx does not have a relation with PVCs.
Genç et al., 2013	Turkey	14/M	1/1	Heart murmur (grade 3–4) over the right inferior sternal border.	Angina and exertional dyspnea.	Yes	Clinical attention.	NA.	RCA.	NA.	Anastomosis LCx and LDA. Fistula to RV.	Damage due to ischemia, EKG shows inferolateral ST depression and T wave changes. The fistula can increase the mortality of the patient if the resolution is not soon.
Giorgio et al., 2010	Italy	9;35/F	2/2	No cardiac disease in FH.	None; dyspnea due to exertion.	Not	Autopsy, Sudden death during exertion.	LCx, LAD.	NA.	NA.	Dominant RCA.	Cause of death was attributed to HCAD, which produces insufficient blood supply to the heart during exertion.
Guo et al., 2012	China	52/M	1/1	Smoker, HTN.	Substernal chest pain, shortness of breath, diaphoresis, and nausea.		Clinical attention.	NA.	LCx.	NA.	RCA large, dominant, 90% of stenosis and narrowing. LAD 90% of stenosis.	It is crucial to recognize this condition when undertaking coronary angiography to determine an appropriate treatment, evermore when the patients are critically ill.
Jariwala et al., 2021	India	12–76/Both	52/52	Not documented.	Acute coronary syndrome, unstable angina, chronic coronary artery syndrome, atypical chest pain, and syncope. Only a few patients were asymptomatic.	Yes	Angiographic or post-mortem findings.	NA.	LCx.	NA.	Dominant RCA.	Temporary ischemia in exertion and chest pain. Important in patients with atherosclerotic disease.
Le Moigne et al., 2022	France	29/M	1/1	Not documented.	Atypical chest pain during exertion and equivocal stress ECG.	Yes	Clinical attention.	NA.	RCA.	NA.	Dominant LCx.	The symptoms were not attributed to the congenital absence of the RCA nor the dominant LCx.
Lin et al., 2003	China	44/F	1/1	Minor Thalassemia.	Exertional chest pain for the last 2 years.	Yes	Clinical attention, Outpatient department.	NA.	LCx.	NA.	Dominant RCA.	Transient ischemia of the inferior and septal wall of the left ventricle during exertion.
Mcfaraland et al., 2011	USA	21/M	1/1	FH for hypertrophy and sudden death.	Dyspnea and increased palpitations, first syncope episode due to exertion.	Yes	Clinical attention.	RCA.	LCx, LDA.	NA.	NA.	It is recommended to perform an ICD placement for patients with HCAD who have experienced syncope episodes.
Morales et al., 2020	USA	37/F	1/1	Born with IUGR, Low birth weight, Dysplastic left multicystic kidney, heart failure.	Dyspnea and desaturation.	Yes	Clinical and Surgical attention and Autopsy due to sudden death.	LMCA.	NA.	NA.	NA.	A treatment with ICD placement or cardiac transplant could have increased the chances of survival.
Nerantzis et al., 2008	Greece	38/M	1/1	Not documented.	Asymptomatic.	Not	Autopsy due to sudden death.	NA.	RCA, LMCA.	LCx separated from LAD origin.	LCx dominant.	Death was not related to anatomy variants.
Oh et al., 2021	Korea	57/M	1/1	Not documented.	Anginal pain that worsened for several hours, squeezing, without irradiation.	Yes	Tertiary care center consultation.	NA.	RCA.	NA.	Dominant LCx.	Complete occlusion of the LCx, resulting in myocardial infarction of the posterior wall of the heart.
Oliveira et al., 2015	Brazil	70/M	1/1	3-month history of exertional angina, 1 episode of syncope after effort, and dyspnea during routine activities.	Syncope and dyspnea during routine activities.	Not	Clinical attention.	NA.	LCx.	LAD arising from the right sinus.	Super-dominant RCA. Aortic stenosis.	The symptoms appeared due to valvular dysfunction. No significant lesions of the coronary arteries.
Phoon et al., 1997	USA	11/M	1/1	Angina and syncope during exertion. No heart disease in FH.	Angina pain, tinnitus, lightheadedness.	Yes	Clinical attention.	NA.	NA.	LMCA arising from right sinus, coursing between the aorta and pulmonary artery.	Arteriovenous malformation of the right atrial branch of the RCA. Stenosis of LMCA.	It was treated with a surgical procedure, and the symptoms were attributed to the stenosis in LMCA.
Pop et al., 2021	Switzerland	53/M	1/1	Current infection with COVID-19.	Angina pain.	Yes	Clinical attention.	NA.	RCA, LMCA.	LCx separated from LAD origin.	LCx dominant. Calcified plaque in LAD.	The patient survived the infection COVID-19. The angina was associated with a calcified plaque in LAD, although it could have been a symptom of the coronary artery variants presented.
Quijada et al., 2014	Spain	51/M	1/1	T2DM, HTN, Obesity.	Asymptomatic.	Not	Clinical attention.	NA.	LCx in its normal parameters.	NA.	RCA dominant, coursing retrogradely. LCx arising as an extension of the posterolateral branch of the right coronary artery, presenting occlusion.	This particular LCx anomaly allows a normal life, even though, can induce angina-like symptoms, particularly during exertion. It is important to underscore a meticulous analysis of these coronary variants, as their symptoms can mimic CAD.
Riede et al., 2013	Germany	16/M	1/1	Athlete, healthy.	Asymptomatic.	Not	Clinical attention in the ICU, and Autopsy due to sudden death caused by H-LCx.	LCx.	NA.	NA.	Dominant LAD.	Coronary angiography enables an accurate detection of coronary abnormalities; therefore, in cases of myocardial infarction in healthy and young patients, is crucial to perform this exam.
Saglam et al., 2017	Turkey	72/F	1/1	Not documented.	Atypical chest pain.	Yes	Clinical attention.	NA.	LMCA, LAD, LCx.	NA.	Single RCA. PDA and PLA supplying the left ventricle.	This extremely rare congenital condition may cause ischemia, acute infarction, syncope, and ventricular fibrillation, and patients are likely to be asymptomatic.
Sánchez et al., 2014	Spain.	102/F	1/1	Not documented.	Cause of death: acute myocardial infarction.	Yes	Anatomical dissection in medical school.	NA	LCx	NA	Super-dominant RCA.	Little-to-no functional repercussions in the course of the donor’s life. No relation with the cause of death.
Sangita et al., 2023	India	25/F	1/1	Obesity, exertional syncope, and dyspnea.	Syncope.	Yes	Autopsy due to sudden death.	LMCA, LAD, LCx.	NA.	NA.	LAD and LCx had no muscular layer.	These anatomical variants led to vessel collapse during exertion, producing myocardial infarction with fatal consequences.
Shaikh et al., 2018	India	48/F	1/1	Not documented.	4-month history of chest pain during exertion.	Yes	Clinical attention	NA	LCx	NA	Super-dominant RCA.	Inducible ischemia showed in a stress test, resulting in angina-like symptoms.
Shi et al., 2021	China	10/M	1/1	Syncope episodes due to exertional activities. No cardiac disease in FH.	Dyspnea, Tachycardia, and Post-exertional syncope, severe angina, upon regaining consciousness.	Yes	Clinical attention.	RCA, LCx, and LAD.	NA.	RCA arises as a trunk of the aorta.	Dominant LMCA.	A genomic sequencing pinpointed a new NOTCH1 c.1023CA site of mutation, resulting in HCAD. This patient was discharged, with restriction to perform any exertional activity, while waiting for a heart transplant.
Smith et al., 1951	USA	60/F	1/1	Not documented.	Severe angina, low heart rate, difficulty breathing, cold, moist, and dusky skin, cyanotic fingernails, and hypoactive reflexes.	Yes	Clinical attention and autopsy due to myocardial infarction.	RCA	NA.	NA.	Arteriosclerosis on both RCA and LMCA and its branches Stenosis and thrombus in LAD.	The authors proposed that H-RCA may have contributed to the large size of the myocardial infarct with rupture of the left ventricle.
Sueza et al., 2019	Japan	50–90/NR	93/5953	Not documented.	Asymptomatic; rest angina.	Not	Clinical attention.	RCA	NA	NA	NA	The authors recommended that in the medical diagnosis of H-RCA with rest angina if spasm provocation tests are not performed, it is advisable to administer a vasodilator, as these patients may have spasm patterns equivalent to triple vessel spasms.
Upadhyaya et al., 2018	UK	47/M	1/1	High BMI, hypertension, and cigarette smoking.	Chest pain with sinus cardiac rhythm.	Yes	Clinical attention	NA	LCA	NA	Large super-dominant RCA.	Cardiac rehabilitation sessions and cardiology clinic follow-up every year.
Utama et al., 2021	Indonesia	37/M	1/1	Situs inversus.	3-month history of chest pain.	Yes	Clinical attention.	NA	LCx	NA	Dominant left-sided RCA	Steal phenomenon triggered by increased requirement of blood in exertion. This results in transient ischemia of the normal, LCx territory.
Widy et al., 2021	Indonesia	37/M	1/1	3-month history of chest pain	Angina.	Yes	Clinical attention.	NA	LCx	NA	Dominant RCA, left sided and LMCA right sided, due to situs inversus condition.	Coronary abnormality symptoms may mimic atherosclerotic cardiac artery disease
Wick et al., 2005	Australia	45/F	1/1	Smoker, HTN.	Asymptomatic.	Not	Autopsy due to sudden death.	RCA	NA	NA	NA	The cause of death was attributed to ischemic heart disease associated with RCA hypoplasia, which produced acute and chronic myocardial ischemic damage.
Yamanaka et al., 1990	USA	9M-82/NR	4/126,595	The study did not include patients with separate origins of the conus branch, right ventricular branch from the right sinus, or coronary anomalies due to complex congenital heart disease.	From asymptomatic to ischemia-like symptoms.	Yes	Coronary arteriography of the patients.	NA	LMCA, LCx.	LCx arising from the right sinus of Valsalva, ectopic origin of RCA, LMCA arising from posterior sinus, CA origin from ascending aorta, ectopic origin from the pulmonary artery.	Intercoronary communication, small coronary artery fistulae.	Patients tend to be asymptomatic. Very few cases can be symptomatic on their own or fatal.
Yan et al., 2018	China	63/F	1/1	5-year history of moderated and atypical chest pain after exertion.	Severe atypical chest.	Yes	Clinical attention.	NA	RCA	NA	Dominant LMCA.	Authors suggested the routine incorporation of CA angiography in order to find cardiovascular variants and to corroborate the result with SCA.
Yoldas et al., 2019	Turkey	14/F	1/1	Not documented.	The cardiac murmur that was found incidentally on routine clinical examination.	Yes	Referred to evaluation due to incidentally detected cardiac murmur.	NA	LMCA	NA	Single RCA with right ventricular fistula.	Asymptomatic patient, correct hemodynamic status of the single RCA.
Yoon et al., 2007	South Korea	48/M	1/1	Hypertension and heavy drinking.	Typical resting chest pain.	Yes	Clinical attention.	NA	LCx	NA	Single LAD and super-dominant RCA.	Vasospastic angina in the single left artery. This angina is not related to the congenital absence of LCx.
Zhou et al., 2023	China	62/F	1/1	Diabetes, hyperlipidemia, and hypertension.	Decreased exercise tolerance and poor blood pressure control.	Yes	Consultation on the cardiology clinic.	NA	LMCA	NA	Single RCA.	Myocardial ischemia is consistent with insufficient blood supply due to the absence of LCA.

**Abbreviations**: EKG/ECG: Electrocardiogram; PVCs: Premature Ventricular Contractions; SCA: Selective Coronary Angiography; H-RCA: Hypoplastic Right Coronary Artery; MDCT: Multidetector Computerized Tomography; CTA: Computerized Tomography Coronary Angiography; SA: Sinoatrial Node; AV: Atrioventricular Node; CAD: Coronary Artery Disease; HCAD: Hypoplastic Coronary Artery Disease, including hypoplasia and congenital absence; HCA: Hypoplastic Coronary Artery; CA: Coronary Artery; RCA: Right Coronary Artery; LMCA: Left Main Coronary Artery; LCx: Left Circumflex Artery; LAD: Left Anterior Descending Artery; ICD: Implantable Cardioverter-Defibrillator.

## Data Availability

The data presented in this study are available on request from the corresponding author. The data are not publicly available due to privacy.

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
