# Peer review of "Hypoplastic and Congenital Absence of Coronary Arteries and Its Correlation with Clinical Implications of Cardiac Circulation: A Systematic Review and Meta-Analysis"

_jcm, 2024, doi:10.3390/jcm13113085_

Round 1
Reviewer 1 Report
Comments and Suggestions for Authors
The authors are conducting a systematic review of congenital coronary artery anomalies for which there is insufficient evidence.
The description of racial differences and anatomical classification of congenital coronary artery anomalies is very interesting.
Major Comments
1. It is difficult to understand what the authors want to clarify in this review.
The first half is redundant and textbook-like, so it should be shortened and rewritten to clarify what is known and what is unknown, so that the reader can understand what part of the unknowns the review focuses on. The background of the abstract should be modified in the same way.
2. The order in which the methods and results are described is not in the same order, making it difficult to read. It would be easier to understand if the outcome in the results section is described in the following order as described in the methods section. i) authors and year of
publication, ii) country, iii) age and sex iv) prevalence v) clinical history vi) circumstances
vii) HCAD viii) congenital absence CA ix) aberrant origin x) other and xi) clinical implications.
3. Please provide data such as the percentage of symptomatic and asymptomatic cases that would provide a basis for concluding that the diagnosis is mostly accidental.
4. Why is there no mention of age at diagnosis?
5. No figure legend is listed. Especially figure 2 and 3 need to be explained for each variant. In addition, it would be helpful for the reader to include a reference paper for each variant.
Please clarify which figure (Figure LⅠ or LⅡA or ?) each of the contents described in the text corresponds to.
Minor comment
1. There are many errors such as spaces and commas. Should be carefully rechecked.
2. The title of figure 3 is incorrect.
Comments on the Quality of English LanguageThere are many errors such as spaces and commas. Should be carefully rechecked.
Author Response
Response reviewer 1
Dear, we appreciate your review and comments, since we are convinced that with the suggested changes our study will improve, below I will detail the response to your proposed comments:
The authors are conducting a systematic review of congenital coronary artery anomalies for which there is insufficient evidence. The description of racial differences and anatomical classification of congenital coronary artery anomalies is very interesting.
Major Comments
- It is difficult to understand what the authors want to clarify in this review. The first half is redundant and textbook-like, so it should be shortened and rewritten to clarify what is known and what is unknown, so that the reader can understand what part of the unknowns the review focuses on. The background of the abstract should be modified in the same way.
R: Dear editor, thank you for your comment, we have considerably summarized the introduction so as not to be redundant and we have focused our results based on what is most important for this review.
- The order in which the methods and results are described is not in the same order, making it difficult to read. It would be easier to understand if the outcome in the results section is described in the following order as described in the methods section. i) authors and year of publication, ii) country, iii) age and sex iv) prevalence v) clinical history vi) circumstances vii) HCAD viii) congenital absence CA ix) aberrant origin x) other and xi) clinical implications.
R: We have ordered the results, so that they are consistent with the methodology, thank you for this comment.
- Please provide data such as the percentage of symptomatic and asymptomatic cases that would provide a basis for concluding that the diagnosis is mostly accidental.
R: We have included whether the diagnosis was accidental or due to symptoms to clarify this criterion, thank you very much.
- Why is there no mention of age at diagnosis?
R: We have added in the results the average year of diagnosis of the participants of the included studies.
- No figure legend is listed. Especially figure 2 and 3 need to be explained for each variant. In addition, it would be helpful for the reader to include a reference paper for each variant. Please clarify which figure (Figure LⅠ or LⅡA or?) each of the contents described in the text corresponds to.
R: We have added the characteristics of the figures in the text, and we have also described them in the figures.
- There are many errors such as spaces and commas. Should be carefully rechecked.
R: We have corrected the entire manuscript to improve the quality of the study.
- The title of figure 3 is incorrect.
R: We have corrected it, thank you very much.

Reviewer 2 Report
Comments and Suggestions for Authors
I had the pleasure of reviewing the paper: „Hypoplastic and congenital absence of coronary arteries and its correlation with clinical implications of cardiac circulation. A systematic review and meta-analysis“ by Bruna-Mejias and associates.
Congenital coronary artery anomalies (CCAA) are rare conditions, and hypoplasia or absence of coronary artery are even less common. Angelini emphasized that a critical attitude in interpreting the “casual vs. causal” relationship in CCAA should be paramount. Accordingly, the authors have targeted exciting and important issues. However, there are several concerns I would like to emphasize just the crucial:
1. First, the writing style and English language need considerable improvements. The assistance from the language expert is warmly recommended.
2. Technically, the references cited throughout the manuscript do not follow the recommended output style (MDPI).
3. Embryological and anatomical descriptions throughout the manuscript are both confusing and missing some up-to-date information (coronary artery embryogenesis).
4. The entire data gathering and pooling for meta-analysis, including the definition of end-point questions, is confusing.
Comments on the Quality of English LanguageWriting style and English language need considerable improvements. The assistance from the language expert is warmly recommended.
Author Response
Response reviewer 2
Dear, we appreciate your review and comments, since we are convinced that with the suggested changes our study will improve, below I will detail the response to your proposed comments:
The authors are conducting a systematic review of congenital coronary artery anomalies for which there is insufficient evidence. The description of racial differences and anatomical classification of congenital coronary artery anomalies is very interesting.
Major Comments
I had the pleasure of reviewing the paper: „Hypoplastic and congenital absence of coronary arteries and its correlation with clinical implications of cardiac circulation. A systematic review and meta-analysis“ by Bruna-Mejias and associates.
Congenital coronary artery anomalies (CCAA) are rare conditions, and hypoplasia or absence of coronary artery are even less common. Angelini emphasized that a critical attitude in interpreting the “casual vs. causal” relationship in CCAA should be paramount. Accordingly, the authors have targeted exciting and important issues. However, there are several concerns I would like to emphasize just the crucial:
- First, the writing style and English language need considerable improvements. The assistance from the language expert is warmly recommended.
R: We have checked the English with a grammar correction service to improve the English in our studio.
- Technically, the references cited throughout the manuscript do not follow the recommended output style (MDPI).
R: We have corrected the references to the journal style, thank you very much
- Embryological and anatomical descriptions throughout the manuscript are both confusing and missing some up-to-date information (coronary artery embryogenesis)
R: We have added in the introduction the embryological characteristics of the heart and the variant
- The entire data gathering and pooling for meta-analysis, including the definition of end-point questions, is confusing.
R: We have corrected and improved this item

Round 2
Reviewer 1 Report
Comments and Suggestions for Authors
Thank you for making the appropriate corrections. I think it is a significant improvement.
However, still needs a few more revisions.
1. The current conclusion statement in the main text was well known before reading this review. It is not the conclusion of this review.
The message I received from this review is as follows:
“Reviewing reports of 811 patients with congenital coronary artery anomalies, most of whom are asymptomatic, with a prevalence of 3% and a mean age of diagnosis of 41 years. Many reports were from Asia, Europe, and North America. Although there are many variants, the characteristic variant should be recognized. The congenital absence of LMCA plus hypoplastic RCA are associated with a higher risk of ischemia, while myocarditis and cardiomyopathy are more likely to manifest in the presence of congenital coronary arteries and are associated with a higher risk of sudden death. Although many reports recommend an ICD for HCAD, the decision should be made on an individual basis depending on the severity of the blood supply due to the degree of collateral vessels development and other factors. It is important for physicians and cardiovascular surgeons to recognize the variants of this anomaly for appropriate management.”
If the author can agree, please rewrite the conclusion after adjusting for excesses and deficiencies with reference to this message.
2. Although the introduction in the text has been adequately revised, background of abstract should also state why this review is necessary.
Minor comment
1. The beginning of 3.6 in result section is a repetition of the same sentence.
Comments on the Quality of English Language“Whilst” in the second paragraph in introduction section is unnecessary.
Author Response
Response reviewer 1
Dear, we appreciate your review and comments, since we are convinced that with the suggested changes our study will improve, below I will detail the response to your proposed comments:
The authors are conducting a systematic review of congenital coronary artery anomalies for which there is insufficient evidence. The description of racial differences and anatomical classification of congenital coronary artery anomalies is very interesting.
Major Comments
- The current conclusion statement in the main text was well known before reading this review. It is not the conclusion of this review.
The message I received from this review is as follows:
“Reviewing reports of 811 patients with congenital coronary artery anomalies, most of whom are asymptomatic, with a prevalence of 3% and a mean age of diagnosis of 41 years. Many reports were from Asia, Europe, and North America. Although there are many variants, the characteristic variant should be recognized. The congenital absence of LMCA plus hypoplastic RCA are associated with a higher risk of ischemia, while myocarditis and cardiomyopathy are more likely to manifest in the presence of congenital coronary arteries and are associated with a higher risk of sudden death. Although many reports recommend an ICD for HCAD, the decision should be made on an individual basis depending on the severity of the blood supply due to the degree of collateral vessels development and other factors. It is important for physicians and cardiovascular surgeons to recognize the variants of this anomaly for appropriate management.”
R: We have improved the conclusion based on your feedback; we appreciate this as we believe we have improved considerably.
- Although the introduction in the text has been adequately revised, background of abstract should also state why this review is necessary.
R: We have added the purpose at the beginning of the review, we appreciate this comment.
Minor comment
- The beginning of 3.6 in result section is a repetition of the same sentence.
R: We have eliminated duplicate content.
Sincerely
Investigation group
